**Data Availability Statement:** The data used in our study was obtained from the Measure DHS program website (http://www.dhsprogram.com),

# Prevalence, spatial distribution and determinants of infant mortality in Ethiopia: Findings from the 2019 Ethiopian Demographic and Health Survey

Tadesse Tarik Tamir[1]*, Tewodros Getaneh Alemu[1], Masresha Asmare Techane[1], Chalachew Adugna Wubneh[1], Nega Tezera Assimamaw[1], Getaneh Mulualem Belay[1], Addis Bilal Muhye[1], Destaye Guadie Kassie[1], Amare Wondim[1], Bewuketu Terefe[2], Bethelihem Tigabu Tarekegn[1], Mohammed Seid Ali[1], Beletech Fentie[1], Almaz Tefera Gonete[1], Berhan Tekeba[1], Selam Fisiha Kassa[1], Bogale Kassahun Desta[1], Amare Demsie Ayele[1], Melkamu Tilahun Dessie[1], Kendalem Asmare Atalell[1]

1 Department of Pediatrics and Child Health Nursing, School of Nursing, College of Medicine and Health Sciences, University of Gondar, Gondar, Ethiopia, 2 Department of Community Health Nursing, School of Nursing, College of Medicine and Health Sciences, University of Gondar, Gondar, Ethiopia

* tadestar140@gmail.com

## Abstract

### Introduction

Infant mortality declined globally in the last three decades. However, it is still a major public health concern in Ethiopia. The burden of infant mortality varies geographically with the highest rate in Sub-Saharan Africa. Although different kinds of literature are available regarding infant mortality in Ethiopia, an up to date information is needed to design strategies against the problem. Thus, this study aimed to determine the prevalence, show the spatial variations and identify determinants of infant mortality in Ethiopia.

### Methods

The prevalence, spatial distribution, and predictors of infant mortality among 5,687 weighted live births were investigated using secondary data from the Ethiopian Demographic and Health Survey 2019. Spatial autocorrelation analysis was used to determine the spatial dependency of infant mortality. The spatial clustering of infant mortality was studied using hotspot analyses. In an unsampled area, ordinary interpolation was employed to forecast infant mortality. A mixed multilevel logistic regression model was used to find determinants of infant mortality. Variables with a p-value less than 0.05 were judged statistically significant and adjusted odds ratios with 95 percent confidence intervals were calculated.

### Result

The prevalence of infant mortality in Ethiopia was 44.5 infant deaths per 1000 live births with significant spatial variations across the country. The highest rate of infant mortality was observed in Eastern, Northwestern, and Southwestern parts of Ethiopia. Maternal age

through formal requests after registering on the website.

**Funding:** The author(s) received no specific funding for this work.

**Competing interests:** The authors have declared that no competing interests exist.

**Abbreviations:** ANC, Antenatal Care; AOR, Adjusted Odds Ratio; CI, Confidence interval; DHS, Demographic and Health Survey; EDHS, Ethiopian Demographic and Health Survey; ICC, Intra-cluster Correlation Coefficient; LLR, Log-Likelihood Ratio; MOR, Median Odds Ratio; PCV, Proportional Change in Variance; SNNPR, Southern Nations Nationalities and People's Region.

between 15&19 (adjusted odds ratio (AOR) = 2.51, 95% Confidence Interval (CI): 1.37, 4.61) and 45&49(AOR = 5.72, 95% CI: 2.81, 11.67), having no antenatal care follow-up (AOR = 1.71, 95% CI: 1.05, 2.79) and Somali region (AOR = 2.78, 95% CI: 1.05, 7.36) were significantly associated with infant mortality in Ethiopia.

## Conclusion

In Ethiopia, infant mortality was higher than the worldwide objective with significant spatial variations. As a result, policy measures and strategies aimed at lowering infant mortality should be devised and strengthened in clustered areas of the country. Special attention should be also given to infants born to mothers in the age groups of 15–19 and 45–49, infants of mothers with no antenatal care checkups, and infants born to mothers living in the Somali region.

## Introduction

Infant mortality is a critical health indicator and reflects a country's overall economic, social, and health systems status [1]. The rate of infant mortality is the number of children who die before reaching the age of one year for every 1000 live births [1]. The prevalence and determinants of infant mortality vary by gender, socioeconomic status, and region [2]. The distribution of scarce resources to areas with the greatest unmet healthcare needs can be improved with the aid of maps showing the variation in infant mortality.

Globally, remarkable improvements had been made to reduce infant mortality from 63 deaths per 1000 live births in 1990 to 29 deaths per 1000 live births in 2016 [3]. However, the progress is not satisfactory in Sub-Saharan Africa, which declined from 182 per 1000 live births in 1990 to 58 deaths per 1000 live births in 2017 [4, 5]. Despite the aim of the Millennium Development Project to reduce infant mortality, the problem remains a challenge in sub-Saharan Africa [6].

In Ethiopia, infant mortality had substantially declined in the last few decades but is behind the global target and persistent regional and local variations [7]. Although the majority of infant deaths are from preventable causes, it has continued to be a complex health problem with many medical, social, and economic determinants [8]. The Ethiopian government has been strongly motivated to improve maternal and child health for decades; however, infant mortality remains a significant health problem in the country [11]. Previous studies identified teenage pregnancy, poor healthcare infrastructures, maternal education, wealth status, Antenatal Care (ANC) utilization, maternal nutrition, place of delivery, child nutrition, size at birth, type of birth, vaccination status, and residence as significant predictors of infant mortality [9–11]. Finding out the current status of factors affecting infant mortality could be an input to have crucial feedback on maternal and child health problems.

Given that infant mortality is the most sensitive health indicator, reflecting both the effectiveness of a nation's healthcare delivery system and its socioeconomic progress in the area of health [12], knowledge of the current prevalence, geographic variation, and risk factors of infant mortality could help public health planners and policymakers develop evidence-based interventions to effectively reduce the problem. As a result, this study aimed to determine prevalence, show spatial distribution, and identify associated factors of infant mortality in Ethiopia using the Ethiopian demographic and health survey (EDHS) 2019.

## Methods

### Study design and setting

Secondary data analysis was done to investigate the prevalence, spatial distribution, and determinants of infant mortality in Ethiopia. Ethiopia is the second and 12[th] most populous country in Africa and worldwide respectively with an average population density of 115 per km2. Administratively Ethiopia has divided into ten regions (Afar, Amhara, Benishangul, Gambela, Harari, Oromia, Somali, Southern, nations, nationalities and people's region (SNNP), Sidama (a recently added), and Tigray) and two city administrations (Addis Ababa and Dire Dawa). More than 84% of the population of Ethiopia resides in rural. Ethiopia uses three-tier health systems: 1) primary care consists of health posts, health centers, and primary hospitals; 2) secondary care consists of zonal hospitals; 3) tertiary care consists of Comprehensive Specialized Hospitals.

### Sample and populations

All babies born from reproductive age women five years preceding the survey in Ethiopia were the source populations for this study, whereas all live births in the selected enumeration areas (EAs) were the study populations. The EDHS uses a two-stage cluster sampling technique. In the first stage, 305 clusters /enumeration areas were randomly selected with a probability sampling stratified with urban and rural. In the second stage, a fixed number of 30 households per cluster were selected using probability sampling.

### Data source

The EDHS 2019 data, which were obtained from the measure demographic and health survey (DHS) through a formal request, were used in this investigation. The measure DHS Program is authorized for the free distribution of unrestricted survey data files for genuine academic study. Registration is a requirement for access to data. The data sets can be downloaded from the website and are freely available to all registered members. The outcome variable for this study was child died before reaching the age of one year. The outcome of the study was dichotomies into "yes" for a child who died before reaching one year and "no" for a child alive beyond one year of age.

Due to the hierarchical nature of the DHS data, the explanatory variables had two-level factors. Individual level factors such as sociodemographic and economic (sex, religion, maternal educational status, maternal age, and wealth index), and the community level factors such as region, residence, level of community poverty, and level of community illiteracy. Geographic coordinate (longitude and latitude) data were also obtained from the EDHS at EAs/cluster level for the spatial analysis.

### Data management and analysis

The data were extracted from the Kids Records (KR) dataset of the EDHS 2019. After cleaning, recoding, the data, and weighting the variables using the women weighing variables, descriptive statistics were carried out and summarized using text, charts, and tables. STATA version 14 and Microsoft Excel 2019 were used for the data management. ArcGIS version 10.8 was used to map infant mortality in Ethiopia. Multicollinearity between factors was checked by using the variation inflation factor (VIF = 3.4), and the issue of possible multiple confounding factors was controlled by using mixed multilevel multivariable logistic regression.

## Spatial analysis

This study used different types of spatial analysis (spatial autocorrelation, hot spot, spatial scan statistical, and interpolation analyses) to show the spatial distribution of infant mortality in Ethiopia. Specifically, spatial autocorrelation was used to determine the presence of spatial clustering [12]. Before hot spot and interpolation analyses, the autocorrelation analysis is a precondition to determine whether the data has spatial dependence. Hot spot analysis identifies the hot spot and cold spot areas of a clustering pattern [12, 13]. Spatial scan statistical analysis identifies the most likely clusters (MLC) (clusters with the set of connected areas that attain the maximum likelihood ratio) [14]. Spatial interpolation is a technique that estimates values in areas where no measured values are available [15].

## Spatial autocorrelations

The possibility of spatial clustering of infant mortality in Ethiopia was determined using spatial autocorrelation (Global Moran's I). Moran's I statistics were used to measure whether infant mortality was distributed randomly, clustered, or dispersed, by taking the entire dataset and producing a single output value, which ranges from -1 to 1. A Moran's I value close to -1 indicates that infant mortality is dispersed in the area. Conversely, Moran's I value, which is closer to 1, shows a localized clustering of infant mortality. In contrast, a value of 0 for Moran's I denote the random distribution of the data. The presence of spatial dependency was indicated by a statistically significant Moran's I ($p < 0.05$).

## Hotspot analysis of infant mortality

Hotspot analysis (Getis-Ord Gi*) was used to determine the spatial clustering of infant mortality in Ethiopia.

## Spatial scan statistical analysis

To locate significant clusters of infant mortality in Ethiopia, spatial scan statistical analysis (Sat Scan) using the Bernoulli model was fitted. The spatial scan statistics use a circular scanning window that moves across the study area. Cases, controls, and geographic coordinate data were fitted to the Bernoulli model. For each potential cluster, Log Likelihood Ratio (LLR), Relative Risk (RR), and P-values were used to determine whether the number of observed cases within the potential cluster was significantly higher than expected or not.

## Spatial interpolation

Using the data in sampled areas, an ordinary Kriging interpolation was performed to forecast infant mortality in unsampled areas.

## Factors associated with infant mortality

A multilevel logistic regression model was fitted to identify the individual and community level factors affecting infant mortality in Ethiopia. Infants are nested within clusters in the EDHS data, and those within the same cluster exhibited greater similarity to one another than those within different clusters. As a result, the independence of observation and equal variance across the cluster assumptions of the ordinary regression model are violated. This suggests a need for use of an advanced model to account for between-cluster factors. Therefore, a multilevel random intercept logistic regression model was fitted to estimate the association between individual-level and community-level factors and the likelihood of infant mortality. Model comparison was done based on deviance (-2log likelihood), the model with the lowest deviance

values was the best-fitted model. Log-likelihood and intracellular correlation coefficient (ICC) was computed to measure the variation between clusters. The ICC indicates the degree of heterogeneity of infant death between clusters.

## Ethics approval and consent to participate

The data used in our study was obtained from the Measure DHS Program website (http://www.dhsprogram.com) through formal requests made after registering on the website. Registration is the only requirement for access to the data. Then the data were available for download within two to three days after registering and requesting them. Registration is the only requirement for access to the data. Therefore, consent to participate was not appropriate since the study was based on secondary data.

## Result

### Sociodemographic characteristics of the study population

A total of 5,687 weighted live births (2,936 males and 2,751 females) were included in the analysis. More than three fourth (77.07%) of the participants were residing in rural areas. Regarding educational status, more than half (54.70%) of the mothers of participants had no formal education. A little bit greater than half (51.38%) of the study participants were born in a poor family (Table 1).

### Prevalence of infant mortality in Ethiopia

The national prevalence of infant mortality rate in Ethiopia was found to be 44.5 deaths per 1000 live births. On the one hand, Afar and Somali regions had high rates of infant mortality. On the other hand, Addis Ababa and Gambela regions of Ethiopia had low rates of infant mortality (Fig 1).

### Spatial analysis of infant mortality in Ethiopia

**Spatial autocorrelation of infant mortality.** According to the findings of our spatial autocorrelation analysis, infant mortality in Ethiopia had a clustering effect, which meant that mortality was high in some areas and low in others. The outputs feature automatically generated keys on the right and left sides of the panel. The z-score value (z = 6.286) of the clustered pattern indicates that the likelihood of it being a random coincidence is less than 1%. The table in the figure demonstrates that the observed value is higher than the predicted values, and the P-value is less than 0.001, indicating statistical significance. This demonstrates that the distribution of infant mortality varies geographically in Ethiopia (Fig 2).

**Hotspot analysis of infant mortality.** In this study, the Local Getis-Ord Gi* statistics was used to identify significant hot spot and cold spot areas of infant mortality. The red and orange color indicates significant hot spot (high-risk) areas for infant mortality and the green color indicates cold spot (low risk) areas. The Northwestern, Southwestern and Eastern parts of Ethiopia were hot spot areas of infant mortality. Cold spots for infant mortality were observed in Northern, Central, and Southern parts of the country (Fig 3).

**Satscan statistical analysis.** Our scan and statistical analysis identified three significant clusters of infant mortality: two primary and one secondary. Among the most likely clusters, two of them were primary clusters, which are located at 11.267438N and 35.292873E with a 57.17km radius. Infants who were born in the primary cluster were 62% more likely to die than those outside the window (relative risk (RR) = 4.62 and log-likelihood ratio (LLR) =

**Table 1. Sociodemographic characteristics of live births born five years preceding the survey in Ethiopia.**

| Variables | Category | Frequency(n) | Percent (%) |
|---|---|---|---|
| Sex | Male | 2,936 | 51.63 |
| | Female | 2,751 | 48.37 |
| Religion | Orthodox | 1,598 | 28.10 |
| | catholic | 32 | 0.56 |
| | protestant | 1,043 | 18.34 |
| | Muslim | 2,932 | 51.56 |
| | Traditional | 62 | 1.09 |
| | other | 20 | 0.35 |
| Residence | Urban | 1,304 | 22.93 |
| Region | Rural | 4,383 | 77.07 |
| | Tigray | 453 | 7.97 |
| | Afar | 629 | 11.06 |
| | Amhara | 506 | 8.90 |
| | Oromia | 716 | 12.59 |
| | Somali | 633 | 11.13 |
| | Benishangul | 528 | 9.28 |
| | SNNPR | 653 | 11.48 |
| | Gambela | 445 | 7.82 |
| | Harari | 444 | 7.81 |
| | Addis Ababa | 288 | 5.06 |
| | Dire Dawa | 392 | 6.89 |
| Maternal education | No education | 3,111 | 54.70 |
| | Primary | 1,801 | 31.67 |
| | Secondary | 475 | 8.35 |
| | Higher | 300 | 5.28 |
| Maternal age | 15–19 | 294 | 5.17 |
| | 20–24 | 1,129 | 19.85 |
| | 25–29 | 1,838 | 32.32 |
| | 30–34 | 1,219 | 21.43 |
| | 35–39 | 757 | 13.31 |
| | 40–44 | 337 | 5.93 |
| | 45–49 | 113 | 1.99 |
| ANC visits | No visits | 302 | 26.77 |
| | <4 visits | 371 | 32.89 |
| | ≥4 visits | 455 | 40.34 |
| Wealth index | Poorest | 1,934 | 34.01 |
| | Poorer | 988 | 17.37 |
| | Middle | 800 | 14.07 |
| | Richer | 735 | 12.92 |
| | Richest | 1,230 | 21.63 |
| Community level of poverty | Low | 587 | 51.22 |
| | High | 559 | 48.78 |
| Community level of illiteracy | Low | 588 | 51.31 |
| | High | 558 | 48.69 |

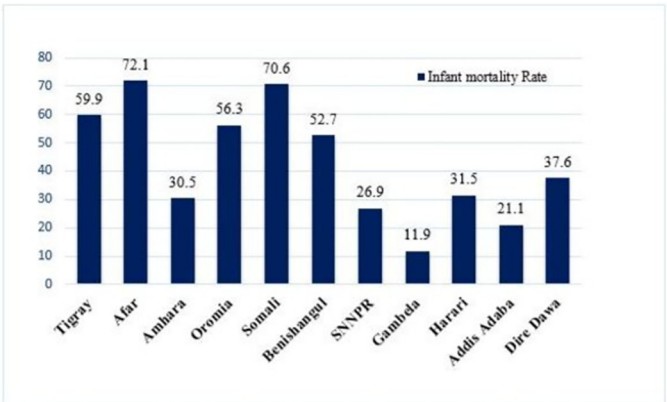

**Fig 1. Regional prevalence of infant mortality in Ethiopia.**

12.16, P-value < 0.001). One of the most likely clusters was a secondary cluster located at 5.069888N and 41.061470E coordinates with a radius of 0 km (Table 2).

**Interpolation of infant mortality.** Ordinary kriging interpolation was used to map the predicted infant mortality in an unsampled area. The high predicted infant mortality was observed in the Eastern, Southwestern, and Northwestern parts of Ethiopia. The Northern, Central, and Southern parts of the country were predicted as areas of low infant mortality (Fig 4).

## Random-effects and model-fit statistics

In the null model, the variance estimate of the random factor is greater than zero ($\tau = 1.08$, p<0.0001) which indicates a significant difference in infant mortality among enumeration

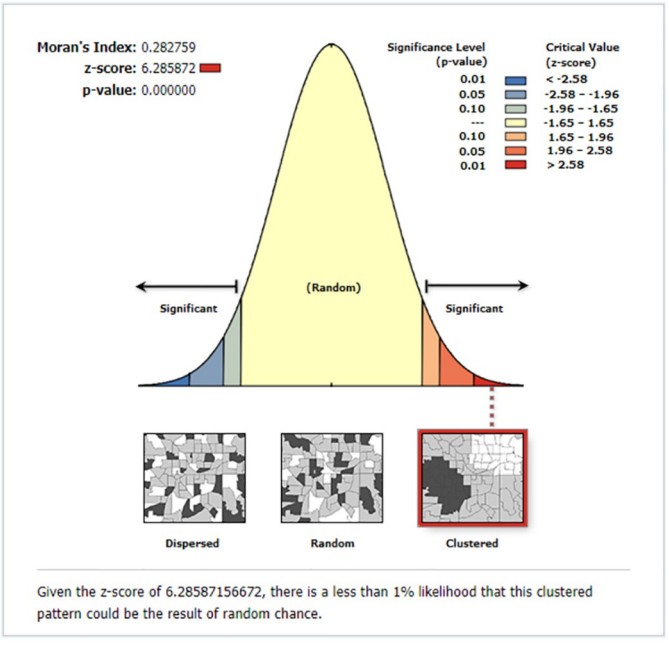

**Fig 2. Spatial autocorrelation of infant mortality in Ethiopia.**

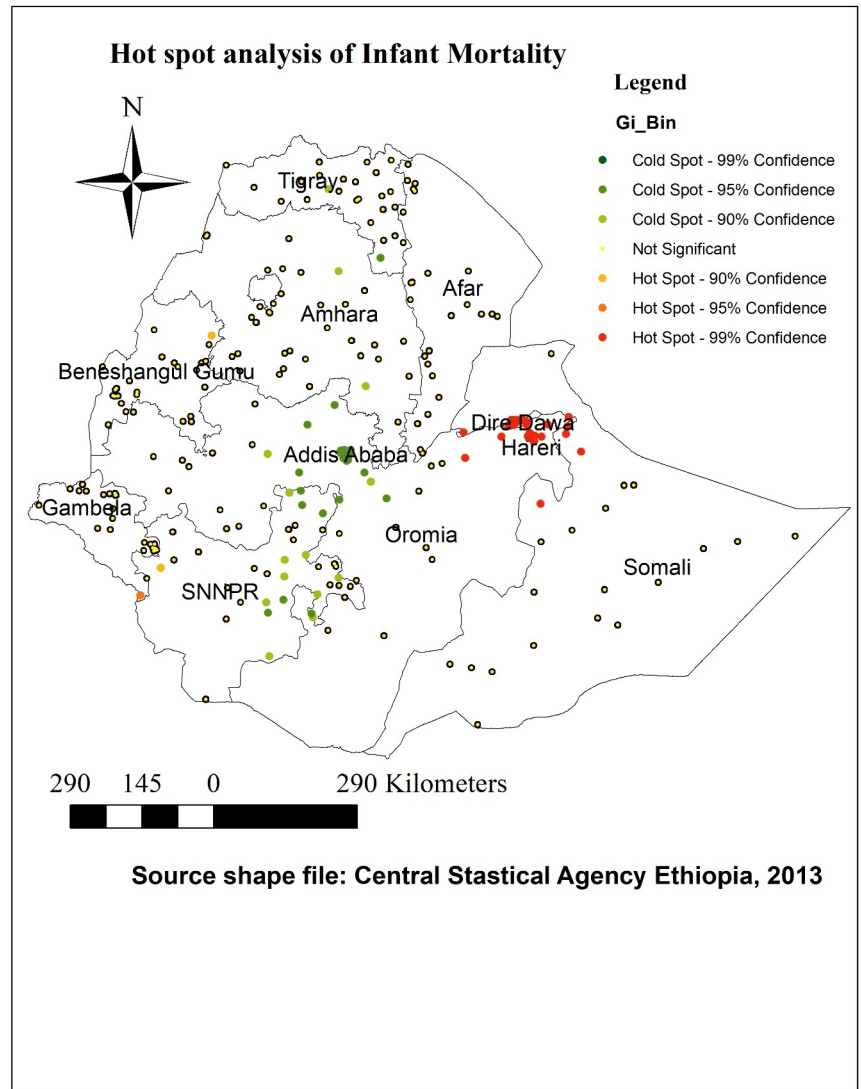

**Fig 3. Hot spot analysis of infant mortality in Ethiopia, Shapefile source: (Central Statistical agency 2013; URL: https://africaopendata.org/dataset/ethiopia-shapefiles).** Map output: Own analysis using ArcMap 10.8 software.

areas (clusters). The ICC in this model indicated that 11% of the total variability of infant death was due to differences across cluster areas, in the remaining unexplained 89% is attributed to individual differences.

Following adjustment of the model for individual-level factors (model II), the variation in the odds of infant mortality remained statistically significant [$\tau = 0.46$, $p < 0.001$] across the communities, with around 58% of the variation in the odds of infant mortality was attributed to individual-level factors (PCV = 57.5%) and 20% of the variation in infant mortality among clusters was attributed to community-level factors (ICC = 20.2%). Model III, which was adjusted for community-level factors showed about 23% variability in odds of infant mortality (PCV = 22.9%) and 20% among cluster variability of infant deaths (ICC = 20.2) was attributed to community-level factors. In the final model (model IV) both the individual level and

**Table 2. Spatial scan statistical analysis of infant mortality in Ethiopia.**

| Cluster | Enumeration areas(cluster) detected | Coordinates/Radius | Population | Cases | RR | LLR | P-value |
|---|---|---|---|---|---|---|---|
| 1 | 159, 160 | (11.267438N, 35.292873 E) /57.17 km | 66 | 15 | 4.62 | 12.16 | <0.001 |
| 2 | 125 | (5.069888N, 41.061470 E) /0 km | 26 | 8 | 6.14 | 8.75 | 0.03 |
| 3 | 281, 282, 284, 283, 287, 285, 286, 288, 296, 291, 297, 292, 294, 290, 293, 289, 295, 302, 303, 304, 305, 298, 301, 299, 300, 108, 251, 231, 232, 253, 236, 246, 238, 237, 240, 233, 243, 235, 242, 239, 234, 241, 244, 252, 247, 245, 254, 249, 255, 248, 250, 107, 109, 127, 121, 130, 128, 106, 43, 129, 126, 88, 40, 50, 133, 105, 42, 28 | (9.618929N, 41.787823E)/189.32 km | 1299 | 95 | 1.63 | 7.60 | 0.08 |
| 4 | 129, 121 | (9.034970 N, 43.060116 E) /47.09 km | 56 | 9 | 3.20 | 4.59 | 0.68 |
| 5 | 304, 305, 303, 296 | (9.514266 N, 41.770584 E) /10.37 km | 101 | 13 | 2.58 | 4.58 | 0.68 |
| 6 | 45 | (12.309340N, 40.271523E) /0 km | 37 | 7 | 3.75 | 4.47 | 0.70 |
| 7 | 210, 223, 222, 224 | (7.229973 N, 35.318649 E) /4.08 km | 37 | 7 | 3.75 | 4.47 | 0.70 |
| 8 | 240, 235, 242, 239, 236, 238, 237 | (9.321629 N, 42.127266 E) /1.95 km | 69 | 10 | 2.89 | 4.33 | 0.76 |
| 9 | 50 | (10.425370 N, 40.353664 E) /0 km | 19 | 4 | 4.15 | 2.91 | 0.10 |

RR, relative risk; LLR, log-likelihood ratio.

community level factors were fitted simultaneously to determine statistically significant factors of infant mortality (Table 3).

**Factors associated with infant mortality in Ethiopia.** The study used deviance to identify the best-fit model due to the nested nature of the model. The model with the lowest deviance value was the best-fitted model. The final best-fitted model, Model IV (deviance = 1254.86) was used for the determination of factors associated with infant mortality.

At the individual level (Model II), maternal age and the number of ANC visits during pregnancy were variables significantly associated with infant mortality. At the community level (Model III), the region was an exceptionally associated variable with infant mortality. Similarly, maternal age, ANC visits, and region remain significant in Model IV.

In multivariable multilevel logistic regression analysis, the odds of mortality among infants born to mothers aged 15–19 and 45–49 years were 2.51(95% CI (1.37, 4.61)) and 5.72 (95% CI (2.81, 11.67)) times higher as compared to those born to mothers aged 25–29 years, respectively. Infants born to mothers having no ANC visits during pregnancy were 1.71(95% CI (1.05, 2.79)) times higher odds of death within the first year of birth than those born to mothers having four or more ANC visits. The odds of infant death were 2.78 (95% CI (1.05, 7.36)) times higher among infants who reside in the Somali region as compared to infants who reside in the Tigray region (Table 4).

## Discussion

Infant mortality which is a critical health indicator; reflects the overall economic, social, and health systems status of a country. This study revealed the prevalence, spatial distribution, and associated factors of infant mortality in Ethiopia using data from EDHS 2019. This offers some insights helpful in tackling the problem in the country.

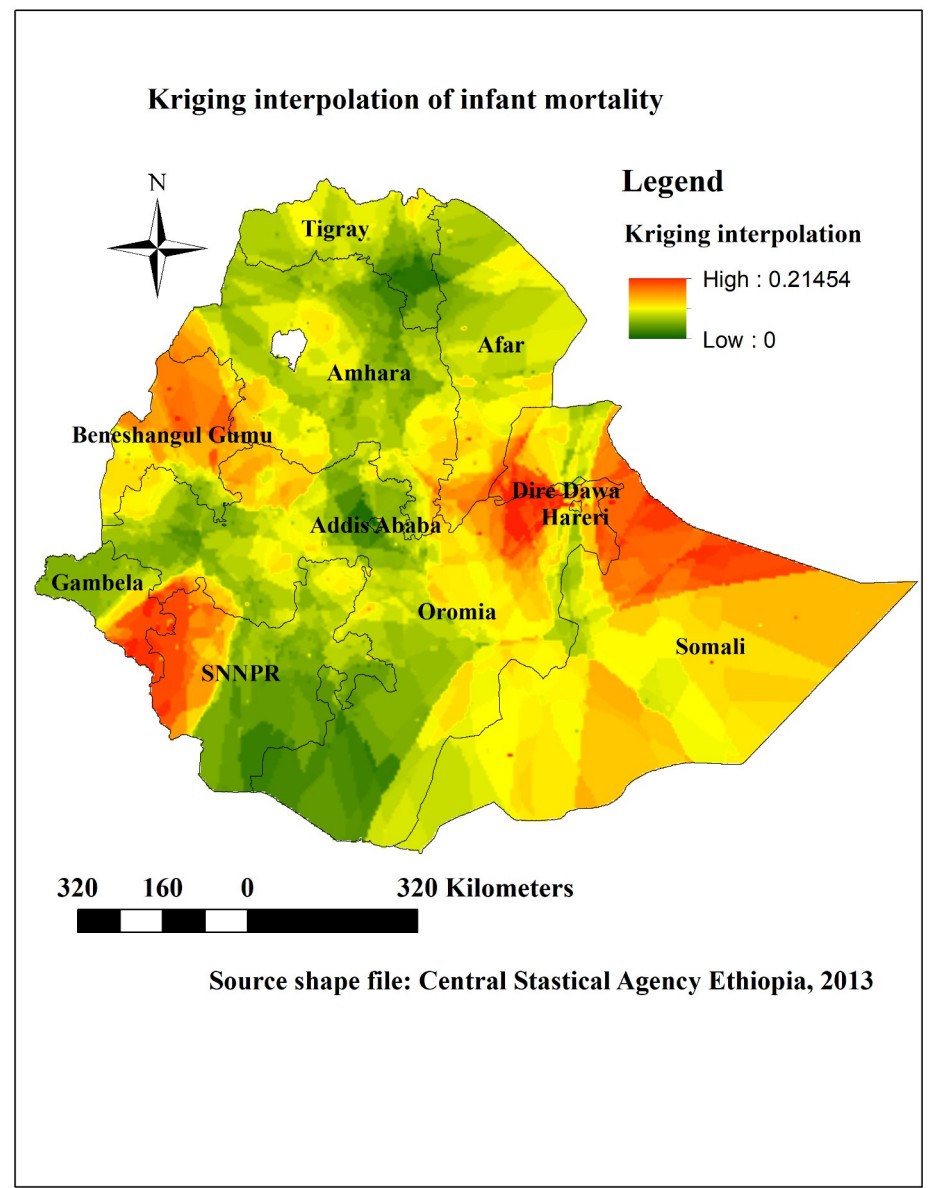

**Fig 4. Interpolation of infant mortality in Ethiopia, Shapefile source: (Central Statistical agency 2013; URL: https://africaopendata.org/dataset/ethiopia-shapefiles). Map output: Own analysis using ArcMap 10.8 software.**

**Table 3. Random effect and model fit statistics of infant mortality in Ethiopia.**

| Parameter | Null model | Model II | Model III | Model IV |
|---|---|---|---|---|
| Variance | 1.08 | 0.46 | 0.83 | 0.42 |
| ICC | 11 | 12 | 20.2 | 11.4 |
| MOR | 2.69 | 1.75 | 2.36 | 1.68 |
| PCV | 1(Reference) | 57.5 | 22.90 | 60.9 |
| Log-likelihood | -1144.39 | -633.79 | -1130.87 | -627.43 |
| Deviance | 2288.79 | 1267.58 | 2261.74 | 1254.86 |

ICC, intracluster correlation coefficient; MOR, Median Odds Ratio; PCV, Proportional Change in Variance.

**Table 4. Multivariable multilevel logistic regression analysis results of both individual-level and community-level factors associated with infant mortality in Ethiopia.**

| Individual and community-level factors | | Model II | Model III | Model IV |
|---|---|---|---|---|
| | | AOR (95%CI) | AOR (95%CI) | AOR (95%CI) |
| sex | Male | 1.33(0.96, 1.85) | | 1.32(0.95, 1.84) |
| | Female | 1 | | 1 |
| Place of delivery | home | 1.17(0.76, 1.78) | | 1.22(0.79, 1.87) |
| | health facility | 1 | | 1 |
| ANC visits | no visits | 1.87(1.17, 3.00)* | | 1.71(1.05, 2.79)* |
| | <4 visits | 0.86(0.55, 1.35) | | 0.86(0.55, 1.36) |
| | ≥4 visits | 1 | | 1 |
| Maternal age | 15–19 | 2.64(1.45, 4.84)* | | 2.51(1.37, 4.61)* |
| | 20–24 | 1.44(0.87, 2.37) | | 1.40(0.85, 2.31) |
| | 25–29 | 1 | | 1 |
| | 30–34 | 0.94(0.54, 1.647) | | 0.96(0.55, 1.69) |
| | 35–39 | 1.56(0.91, 2.68) | | 1.63(0.94, 2.81) |
| | 40–44 | 1.86(0.96, 3.612) | | 1.89(0.97, 3.66) |
| | 45–49 | 5.76(2.84, 11.67)* | | 5.72(2.81, 11.67)* |
| Maternal educational status | no education | 2.55(0.75, 8.70) | | 2.65(0.77, 9.11) |
| | primary | 3.05(0.92, 10.12) | | 3.21(0.96, 10.71) |
| | Secondary | 1.76(0.46, 6.68) | | 1.87(0.49, 7.08) |
| | higher | 1 | | 1 |
| Wealth index | poorest | 0.74(0.43, 1.29) | | 0.68(0.29, 1.554) |
| | Poorer | 0.62(0.34, 1.13) | | 0.66(0.29, 1.50) |
| | middle | 0.70(0.38, 1.29) | | 0.79(0.36, 1.73) |
| | richer | 0.94(0.53, 1.66) | | 1.09(0.53, 2.23) |
| | Richest | 1 | | 1 |
| Region | Tigray | | 1 | 1 |
| | Afar | | 1.97(0.90, 4.33) | 1.83(0.68, 4.94) |
| | Amhara | | 1.59(0.71, 3.52) | 1.58(0.61, 4.14) |
| | Oromia | | 2.08(0.99, 4.33) | 1.73(0.70 4.28) |
| | Somali | | 2.78(1.28, 6.05)* | 2.78(1.05, 7.36)* |
| | Benishangul | | 2.93(1.38, 6.20)* | 2.17(0.86, 5.51) |
| | SNNPR | | 1.14(0.51, 2.52) | 0.91(0.33, 2.48) |
| | Gambela | | 1.98(0.89, 4.41) | 1.76(0.66, 4.71) |
| | Harari | | 2.43(1.10, 5.36)* | 1.60(0.57, 4.46) |
| | Addis Ababa | | 0.76(0.25, 2.275) | 1.32(0.39, 4.43) |
| | Dire Dawa | | 2.46(1.11, 5.49) | 2.36(0.89, 6.28) |
| Residence | Urban | | 1 | 1 |
| | Rural | | 0.94(0.61, 1.43) | 0.91(0.51, 1.64) |
| Community poverty level | Low | | 1 | 1 |
| | High | | 1.15(0.80, 1.65) | 1.16(0.69, 1.945) |
| Community women illiteracy level | Low | | 1 | 1 |
| | High | | 0.98(0.68, 1.40) | 0.84(0.53, 1.33) |

SNNPR, Southern Nations Nationalities and Peoples Region;

*, P <0.05(significantly associated).

The national prevalence of infant mortality in Ethiopia was 44.5 infant deaths per 1000 live births (95% CI: 39, 50). This finding was consistent with a previous study conducted in rural areas of Ethiopia with 47 infant deaths per 1000 live births [16]. It was also in line with the national infant mortality rate (IMR) of some East African countries; Tanzania 51 per 1000 live births, Uganda 45 per 1000 live births, and Kenya 49 per 1000 live births [17, 18]. Nonetheless, the finding was higher than the study conducted in the United States by 3.05% [19]. This could be due to differences in health policy, quality of health care, socioeconomic and cultural differences across the countries. Our findings' higher prevalence suggests the necessity for targeted solutions to the issue.

The spatial distribution of infant mortality in Ethiopia was non-random. This was supported by previous studies [20–24]. The spatial hot spot analysis indicated that geographic variations in infant mortality were observed in the Northern, Northwestern, and Eastern parts of Ethiopia. In addition, the prediction of high infant mortality was observed from spatial interpolation analysis in the Eastern, Southwestern and Northwestern parts of Ethiopia. The possible explanation for such a variation in the distribution of infant mortality may be the varied distributions in etiologies and risk factors of infant mortality from one part of the country to other parts. Evidence also shows that the limited accessibility of maternal and child health care services such as ANC, health facility delivery, Post Natal Care(PNC), and childhood vaccinations in the border areas of Ethiopia could have resulted in variability in the distribution of infant mortality in the country [20]. This suggests that locations, where infant death is more prevalent, require the application of unique strategies and interventions to improve health and increase the survival chances of infants.

The results of our multilevel analysis of EDHS 2019 identified the significant factors that are associated with infant mortality in Ethiopia. In the multivariable multilevel logistic regression model; maternal age, the number of ANC visits during pregnancy, and region were found to be significant predictors of infant mortality.

In our study, maternal age was significantly associated with infant mortality. The odds of infant mortality among births from mothers aged 15–19 years were higher than births from mothers aged 25–29 years. This is supported by previous studies [25]. This could be due to the reason that teenagers are not biologically mature enough to properly care for and nourish their babies which could raise the risk of infant death [26]. In addition, teenagers are less likely to use maternal health care services such as ANC, institutional delivery, postnatal care, and routine immunization this could cause the odds of infant mortality to increase. The births from mothers aged 45–49 years were significantly associated with the mortality rate of infants using births from mothers aged 25–29 years as a reference group. The previous study has witnessed the same [27]. These results in our findings indicate that babies born to mothers below and above the peak of their reproductive capacity are more likely to die than babies born to mothers who are not. To lower infant mortality in Ethiopia, maternal and child health programs should take babies born to mothers in extreme reproductive life into consideration.

Regarding the number of ANC visits during pregnancy as a significantly associated factor of infant mortality, the odds of infant mortality among infants born to mothers who had no ANC visit during pregnancy were higher when compared to those who have four and above ANC visits. This was supported by other studies [21, 28, 29]. It could be due to the reason that ANC utilization is an input for the other maternal health services, and mothers who had no ANC contact with a skilled health professional are not aware of danger signs of pregnancy and underlying medical conditions that could cause low birth weight, prematurity and congenital anomalies which may intern lead to the death of their infant [30].

Finally, the region was a community-level factor significantly associated with infant mortality in Ethiopia. Works of literature also supported this regional variation in infant mortality

[31]. Infants born to mothers who live in the Somali region were more likely to die within one year of age compared to those who reside in the Tigray region. The possible explanation for this could be residents in the Somali region relatively have low access to health infrastructure, education, and informational resources in comparison to Tigray [32]. The finding could also be explained by the fact that the Somali region experiences extremely high temperatures throughout the year, both during the day and less cooling overnight [33]. This has an impact on the health of humans, especially that of small children, and may contribute to an increase in infant mortality in the area.

To reduce the impact of infant mortality on children's survival and growth rates in Ethiopia, policymakers should develop infant mortality prevention measures that take into account prevalence, distribution, and determinants.

## Strength and limitations

One strength of this study was the use of large, nationally representative sample data. Another virtue of the study was the use of mixed multilevel logistic regression to determine two-level factors, which could not be done using ordinary logistic regression. However, due to the use of secondary data, the study was limited in its ability to include other variables that might have been associated with the outcome variable. Since the study used spatial analysis, it may be subjected to spatial bias like map area and projection bias.

## Conclusion

The rate of infant mortality in Ethiopia was higher than the global target, with significant spatial variations throughout the country. Maternal age, ANC follow-up, and being a resident of the Somali region were identified as determinants of infant mortality. Thus, policy measures and strategies aimed at lowering infant mortality should be devised and strengthened in clustered areas of the country. Special attention should be also given to infants born to mothers in the age groups of 15–19 and 45–49, infants of mothers with no antenatal care checkups, and infants born to mothers living in the Somali region.

## Author Contributions

**Conceptualization:** Tewodros Getaneh Alemu, Chalachew Adugna Wubneh, Getaneh Mulualem Belay, Bewuketu Terefe, Mohammed Seid Ali, Berhan Tekeba, Selam Fisiha Kassa.

**Data curation:** Chalachew Adugna Wubneh, Bewuketu Terefe, Mohammed Seid Ali, Beletech Fentie, Berhan Tekeba, Selam Fisiha Kassa, Amare Demsie Ayele.

**Formal analysis:** Tadesse Tarik Tamir, Melkamu Tilahun Dessie.

**Investigation:** Amare Wondim, Beletech Fentie.

**Methodology:** Tadesse Tarik Tamir, Addis Bilal Muhye, Almaz Tefera Gonete, Bogale Kassahun Desta.

**Software:** Tadesse Tarik Tamir.

**Supervision:** Nega Tezera Assimamaw, Amare Demsie Ayele.

**Validation:** Amare Wondim, Selam Fisiha Kassa, Kendalem Asmare Atalell.

**Visualization:** Masresha Asmare Techane, Destaye Guadie Kassie, Amare Wondim, Bethelihem Tigabu Tarekegn, Mohammed Seid Ali, Almaz Tefera Gonete, Selam Fisiha Kassa, Melkamu Tilahun Dessie, Kendalem Asmare Atalell.

**Writing – original draft:** Tadesse Tarik Tamir.

**Writing – review & editing:** Tewodros Getaneh Alemu, Masresha Asmare Techane, Nega Tezera Assimamaw, Getaneh Mulualem Belay, Destaye Guadie Kassie, Bethelihem Tigabu Tarekegn, Berhan Tekeba, Bogale Kassahun Desta, Amare Demsie Ayele, Kendalem Asmare Atalell.

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
