## [Decision Letter · Decision Letter 0]

20 Sep 2022

PONE-D-22-21335Prevalence, spatial distribution and determinants of Infant Mortality in Ethiopia: Findings from 2019 Ethiopian Demographic and Health SurveyPLOS ONE

Dear Dr. Tamir,

Thank you for submitting your manuscript to PLOS ONE. After careful consideration, we feel that it has merit but does not fully meet PLOS ONE’s publication criteria as it currently stands. Therefore, we invite you to submit a revised version of the manuscript that addresses the points raised during the review process.

Please submit your revised manuscript by Nov 04 2022 11:59PM. If you will need more time than this to complete your revisions, please reply to this message or contact the journal office at plosone@plos.org. Please include the following items when submitting your revised manuscript:A rebuttal letter that responds to each point raised by the academic editor and reviewer(s). You should upload this letter as a separate file labeled 'Response to Reviewers'.A marked-up copy of your manuscript that highlights changes made to the original version. You should upload this as a separate file labeled 'Revised Manuscript with Track Changes'.An unmarked version of your revised paper without tracked changes. You should upload this as a separate file labeled 'Manuscript'.

We look forward to receiving your revised manuscript.

Kind regards,

Betregiorgis Zegeye

Academic Editor

PLOS ONE

Journal Requirements:

2. Thank you for submitting the above manuscript to PLOS ONE. During our internal evaluation of the manuscript, we found significant text overlap between your submission and previous work in the Methods section. We would like to make you aware that copying extracts from previous publications word-for-word is unacceptable. In addition, the reproduction of text from published reports has implications for the copyright that may apply to the publications. Please revise the manuscript to rephrase the duplicated text, cite your sources, and provide details as to how the current manuscript advances on previous work. Please note that further consideration is dependent on the submission of a manuscript that addresses these concerns about the overlap in text with published work. We will carefully review your manuscript upon resubmission and further consideration of the manuscript is dependent on the text overlap being addressed in full. Please ensure that your revision is thorough as failure to address the concerns to our satisfaction may result in your submission not being considered further.

5. We note that Figures 3 and 4 in your submission contain map images which may be copyrighted. All PLOS content is published under the Creative Commons Attribution License (CC BY 4.0), which means that the manuscript, images, and Supporting Information files will be freely available online, and any third party is permitted to access, download, copy, distribute, and use these materials in any way, even commercially, with proper attribution. For these reasons, we cannot publish previously copyrighted maps or satellite images created using proprietary data, such as Google software (Google Maps, Street View, and Earth). For more information, see our copyright guidelines: http://journals.plos.org/plosone/s/licenses-and-copyright.

a. You may seek permission from the original copyright holder of Figures 3 and 4 to publish the content specifically under the CC BY 4.0 license.  

Reviewers' comments:

Reviewer's Responses to Questions

**Comments to the Author**

1. Is the manuscript technically sound, and do the data support the conclusions?

Reviewer #1: No

Reviewer #2: Yes

2. Has the statistical analysis been performed appropriately and rigorously? 

Reviewer #1: I Don't Know

Reviewer #2: I Don't Know

3. Have the authors made all data underlying the findings in their manuscript fully available?

Reviewer #1: Yes

Reviewer #2: Yes

4. Is the manuscript presented in an intelligible fashion and written in standard English?

Reviewer #1: Yes

Reviewer #2: No

5. Review Comments to the Author

Reviewer #1: The paper has issues in the presentation of the findings and linking the spatial analysis with the regression analysis to which I have identified comments, and I encourage the authors to address the substantive nature of these comments major however, considering the importance of the findings of the paper to understanding IMR in Ethiopia, I ask the authors to revisit the paper to address my comments:

Reviewer #2: REVIEWER COMMENTS

To Publisher, PLOSONE

Sept 10,2022

manuscript ID: PONE-D-22-21335

Title: Prevalence, spatial distribution, and determinants of Infant Mortality in Ethiopia: Findings from 2019 Ethiopian Demographic and Health Survey

Thank you for giving me the opportunity to review the manuscript. Here below I have attached comments here

In ABSTRACT, in the conclusion subsection, the author stated that PNC coverage in clustered regions should be devised and strengthened. However, in their study, PNC coverage was not a significant predictor of infant mortality. The authors should emphasize what they found significant in their study. Moreover. The abbreviation PNC has not been defined elsewhere in this section; therefore, it is not clear what it represents. The statement was higher than the worldwide objective and is not clear. What is the objective worldwide? An appropriate term should be a goal or a target.

The abbreviations should be expanded when used for the first time. (eg, PNC, EDHS, DHS).

In result,

¨A little beat greater than half (51.38%) of the study participants were born from poor family (Table 1) ¨. Beat should be replaced with bit; however, the statement was not written in a formal way to write the article.

The discussion section is inadequate.

In addition, there are several language and editorial issues.

Thank you

6. PLOS authors have the option to publish the peer review history of their article (what does this mean?). If published, this will include your full peer review and any attached files.

Reviewer #1: **Yes: **Dr Danish Ahmad

Reviewer #2: No

---

## [Author Response · Author response to Decision Letter 0]

8 Nov 2022

Dear reviewers, thank you for an insightful comments, suggestions and guidance which significantly improved our manuscript.

---

## [Decision Letter · Decision Letter 1]

31 Jan 2023

PONE-D-22-21335R1Prevalence, spatial distribution and determinants of Infant Mortality in Ethiopia: Findings from 2019 Ethiopian Demographic and Health SurveyPLOS ONE

Dear Dr. Tadesse,

Thank you for submitting your manuscript to PLOS ONE. After careful consideration, we feel that it has merit but does not fully meet PLOS ONE’s publication criteria as it currently stands. Therefore, we invite you to submit a revised version of the manuscript that addresses the points raised during the review process.

Please submit your revised manuscript by Mar 17 2023 11:59PMIf you will need more time than this to complete your revisions, please reply to this message or contact the journal office at plosone@plos.org. Please include the following items when submitting your revised manuscript:A rebuttal letter that responds to each point raised by the academic editor and reviewer(s). You should upload this letter as a separate file labeled 'Response to Reviewers'.A marked-up copy of your manuscript that highlights changes made to the original version. You should upload this as a separate file labeled 'Revised Manuscript with Track Changes'.An unmarked version of your revised paper without tracked changes. You should upload this as a separate file labeled 'Manuscript'.If applicable, we recommend that you deposit your laboratory protocols in protocols.io to enhance the reproducibility of your results. Protocols.io assigns your protocol its own identifier (DOI) so that it can be cited independently in the future. For instructions see: https://journals.plos.org/plosone/s/submission-guidelines#loc-laboratory-protocols. Additionally, PLOS ONE offers an option for publishing peer-reviewed Lab Protocol articles, which describe protocols hosted on protocols.io. Read more information on sharing protocols at https://plos.org/protocols?utm_medium=editorial-email&utm_source=authorletters&utm_campaign=protocols.

We look forward to receiving your revised manuscript.

Kind regards,

Betregiorgis Zegeye,

Academic Editor

PLOS ONE

Journal Requirements:

Additional Editor Comments

Please be sure that each of the comment raised by reviewer is addressed before you submit it and also it need repeated proofing for language editing.

Reviewers' comments:

Reviewer's Responses to Questions

**Comments to the Author**

1. If the authors have adequately addressed your comments raised in a previous round of review and you feel that this manuscript is now acceptable for publication, you may indicate that here to bypass the “Comments to the Author” section, enter your conflict of interest statement in the “Confidential to Editor” section, and submit your "Accept" recommendation.

Reviewer #3: All comments have been addressed

Reviewer #4: All comments have been addressed

2. Is the manuscript technically sound, and do the data support the conclusions?

Reviewer #3: Yes

Reviewer #4: Yes

3. Has the statistical analysis been performed appropriately and rigorously? 

Reviewer #3: Yes

Reviewer #4: Yes

4. Have the authors made all data underlying the findings in their manuscript fully available?

Reviewer #3: Yes

Reviewer #4: Yes

5. Is the manuscript presented in an intelligible fashion and written in standard English?

Reviewer #3: Yes

Reviewer #4: Yes

6. Review Comments to the Author

Reviewer #3: Thank you for addressing the points which were mentioned in the previous review. The responses are satisfactory and clarify the points previously highlighted.

Reviewer #4: 1. As you know the abbreviations that come for the first time in the text should be stated completely. (ANC and PNC in abstract).

2. Are there any exclusion criteria for this survey?

3.I Recommend to use the recent literature in disscution.

4. According to the findings, it would be better that you make some recommendations at the end of discussion.

5. I recommend to clarify the findings in figure 2 in the result part, such as critical values.

6. The map and graphs are well designed.

7. PLOS authors have the option to publish the peer review history of their article (what does this mean?). If published, this will include your full peer review and any attached files.

Reviewer #3: No

Reviewer #4: **Yes: **Mehrandokht Abedini

---

## [Author Response · Author response to Decision Letter 1]

7 Feb 2023

Response to comments

PONE-D-22-21335

Prevalence, spatial distribution and determinants of Infant Mortality in Ethiopia: Findings from 2019 Ethiopian Demographic and Health Survey.

Journal: PLOS ONE 

Subject: Submission of revised manuscript

We sincerely appreciate the editor's and reviewer's comments on this work. Kindly find our response to the comments below. Please refer to the attached file, which was uploaded independently and contains the revised manuscript.

Yours sincerely, 

Tadesse Tarik Tamir (on behalf of all authors)

University of Gondar, Gondar, Ethiopia

Response to Editor comments

Response: Dear editor, thank you for letting us know the PLOS ONE requirement regarding the reference list. We have reviewed our references and we have not used retracted references. 

Response: Dear editor, thank you for your suggestions. We thoroughly made corrections to typographical or grammatical errors in the manuscript.

Response to reviewer comments 

Reviewer #4

Dear reviewer, First and foremost, we appreciate your enthusiasm for our manuscript's subject and hypotheses, as well as your detailed perspectives and insightful comments.

1. As you know the abbreviations that come for the first time in the text should be stated completely. (ANC and PNC in abstract).

Response: Dear reviewer, you are correct abbreviations that come for the first time in the text should be stated completely. The concern is addressed accordingly. 

2. Are there any exclusion criteria for this survey?

Response: Dear reviewer, thank you for the concern. For the survey on which our analysis was done, there are no declared exclusion criteria that could have an effect on our results.

3.I Recommend to use the recent literature in disscution.

Response: Dear reviewer, thank you for such a recommendation that makes our manuscript scientifically sound. We replaced some references in the discussion with more recent ones as per your recommendation. We used more recent references in our manuscript if they were otherwise not available with the best of our search.

4. According to the findings, it would be better that you make some recommendations at the end of discussion.

Response: Dear reviewer, thank you for such an insightful comment. We made a kind of recommendation at the end of the discussion as per your direction. Please find the point at the end of the discussion in our revised manuscript. 

5. I recommend to clarify the findings in figure 2 in the result part, such as critical values.

Response: Dear reviewer, we made modifications to the section as per your recommendation. Kindly find the point in the result part of the revised manuscript. 

6. The map and graphs are well designed.

Response: Dear reviewer, we are overjoyed to have such positive feedback on our work from such esteemed reviewers as you.

---

## [Editor Report · Decision Letter 2]

10 Apr 2023

Prevalence, spatial distribution and determinants of Infant Mortality in Ethiopia: Findings from 2019 Ethiopian Demographic and Health Survey

PONE-D-22-21335R2

Dear Dr. Tadesse,

We’re pleased to inform you that your manuscript has been judged scientifically suitable for publication and will be formally accepted for publication once it meets all outstanding technical requirements.

Kind regards,

Betregiorgis Zegeye, 

Academic Editor

PLOS ONE

---

## [Editor Report · Acceptance letter]

14 Apr 2023

PONE-D-22-21335R2 

Prevalence, spatial distribution and determinants of Infant Mortality in Ethiopia: Findings from the 2019 Ethiopian Demographic and Health Survey 

Dear Dr. Tamir:

I'm pleased to inform you that your manuscript has been deemed suitable for publication in PLOS ONE. Congratulations! Your manuscript is now with our production department. 

Kind regards, 

on behalf of

Mr. Betregiorgis Hailu Zegeye 

Academic Editor

PLOS ONE